# Structure and flexibility of the yeast NuA4 histone acetyltransferase complex

Stefan A Zukin[1†], Matthew R Marunde[2], Irina K Popova[2], Katarzyna M Soczek[3,4,5], Eva Nogales[3,4,6,7]*, Avinash B Patel[3,7,8]*‡

[1]College of Chemistry, University of California, Berkeley, Berkeley, United States; [2]EpiCypher, Inc, Research Triangle Park, Durham, United States; [3]California Institute for Quantitative Biology, University of California, Berkeley, Berkeley, United States; [4]Department of Molecular and Cellular Biology, University of California, Berkeley, United States; [5]Innovative Genomics Institute, University of California, Berkeley, Berkeley, United States; [6]Molecular Biophysics and Integrative Bio-Imaging Division, Lawrence Berkeley National Laboratory, Berkeley, United States; [7]Howard Hughes Medical Institute, University of California, Berkeley, Berkeley, United States; [8]Biophysics Graduate Group, University of California, Berkeley, Berkeley, United States

*For correspondence:
enogales@lbl.gov (EN);
avinash.patel@northwestern.edu
(ABP)

Present address: †Department of Genetics, Harvard Medical School, Boston, United States; ‡Molecular Biosciences, Northwestern University, Evanston, United States

**Abstract** The NuA4 protein complex acetylates histones H4 and H2A to activate both transcription and DNA repair. We report the 3.1-Å resolution cryo-electron microscopy structure of the central hub of NuA4, which flexibly tethers the histone acetyltransferase (HAT) and Trimer Independent of NuA4 involved in Transcription Interactions with Nucleosomes (TINTIN) modules. The hub contains the large Tra1 subunit and a core that includes Swc4, Arp4, Act1, Eaf1, and the C-terminal region of Epl1. Eaf1 stands out as the primary scaffolding factor that interacts with the Tra1, Swc4, and Epl1 subunits and contributes the conserved HSA helix to the Arp module. Using nucleosome-binding assays, we find that the HAT module, which is anchored to the core through Epl1, recognizes H3K4me3 nucleosomes with hyperacetylated H3 tails, while the TINTIN module, anchored to the core via Eaf1, recognizes nucleosomes that have hyperacetylated H2A and H4 tails. Together with the known interaction of Tra1 with site-specific transcription factors, our data suggest a model in which Tra1 recruits NuA4 to specific genomic sites then allowing the flexible HAT and TINTIN modules to select nearby nucleosomes for acetylation.

## Editor's evaluation

This manuscript provides insights into the architecture of the yeast histone acetyltransferase complex NuA4 and is of broad interest to those studying transcription and chromatin modification. The cryo-EM data are of very high quality, and enable the authors to devise a structural model that is in much better agreement with biochemical data than previously reported models. This structure represents an important puzzle piece towards a molecular understanding of chromatin modification.

## Introduction

Chemical modifications of histones are a key mechanism by which gene expression is regulated. These chemical modifications can affect the physical state of chromatin, regulating whether it is in a tightly packed and repressed state, or in an open and active state (*Jenuwein and Allis, 2001*; *Strahl and Allis, 2000*; *van Attikum and Gasser, 2005*; *Grunstein, 1997*). One such chemical modification, lysine acetylation, is catalyzed by histone acetyltransferases (HATs), which are often part of large,

multisubunit complexes (*Carrozza et al., 2003*; *Lee and Workman, 2007*). Acetylated histones can directly affect the stability of a nucleosome by neutralizing the otherwise positively charged lysines tails that help stabilize the binding of the octamer core to the negatively charged DNA, or indirectly by recruiting chromatin remodelers that can alter how histone octamers bind DNA (*Swygert and Peterson, 2014*; *Workman and Kingston, 1998*; *Morrison et al., 2018*; *Brower-Toland et al., 2005*).

NuA4 (Nucleosome Acetyltransferase of H4) is one of eight HAT-containing complexes in *Saccharomyces cerevisiae*, and its catalytic subunit, Esa1 (or Kat5), is the only essential HAT in *S. cerevisiae* (*Lee and Workman, 2007*; *Grant et al., 1997*; *Eberharter et al., 1998*; *MacDonald and Howe, 2009*; *Smith et al., 1998*; *Boudreault et al., 2003*; *Steunou et al., 2014*; *Allard et al., 1999*; *Allis et al., 2007*). NuA4 is composed of a total of 13 subunits, which together give the complex a molecular weight of about 1 MDa (*Boudreault et al., 2003*). The complex is thought to be organized in four main parts: the Tra1 subunit, the core module, the HAT module, and the Trimer Independent of NuA4 involved in Transcription Interactions with Nucleosomes (TINTIN) module (*Doyon and Côté, 2004*). Tra1, the largest component of NuA4, is a member of the phosphoinositide-3-kinase (PI3K)-related pseudo-kinase (ΨPIKK) family of proteins, which lacks kinase activity and instead functions as the primary target for the binding of sequence-specific transcription factors (*Allard et al., 1999*; *McMahon et al., 1998*; *Brown et al., 2001*; *Grant et al., 1998*). The core module has been proposed to include Eaf1, Swc4, Yaf9, Arp4, and Act1, and is thought to connect the other three parts of the complex together (*Boudreault et al., 2003*; *Allard et al., 1999*; *Doyon and Côté, 2004*). The HAT module (also known as Piccolo) contains Esa1, Yng2, Eaf6, and Epl1 and is responsible for the acetylation of histone H2A and H4 in target nucleosomes (*Boudreault et al., 2003*). And lastly the TINTIN module is composed of Eaf3, Eaf5, and Eaf7, and has several speculated functions including binding the Pol II CTD, RNA, and histones (*Bhat et al., 2015*).

A previously reported cryo-electron microscopy (cryo-EM) structure of NuA4 (*Wang et al., 2018*) proposed that the Tra1 subunit and core module forms a rigid connection, while the HAT module appears more flexibly attached. However, crosslinking mass spectrometry data were incompatible with much of the de novo built regions of the structure or with the location of the HAT module proposed in later negative stain studies (*Wang et al., 2018*; *Setiaputra et al., 2018*).

Here, we have used cryo-EM to visualize NuA4 and have resolved the stable central hub containing the Tra1 subunit and the core module at ~3-Å resolution, allowing for accurate subunit assignments to all the density within this region. We found that the core is composed of Eaf1, Swc4, Arp4, and Act1 as well as the C-terminus of Epl1. The flexible HAT and TINTIN modules are tethered to the core through Epl1 and Eaf1, respectively. Using nucleosome-binding assays, we were able to show that the HAT module prefers nucleosomes modified with H3K4me3 and hyperacetylated H3 tails (acetylation marks that are produced by the SAGA complex), while the TINTIN module has a weak preference for nucleosomes that are hyperacetylated at their H2A and H4 tails (the product of the NuA4 HAT module). Based on our findings, we propose a model of how NuA4 and other HAT complexes target nucleosomes for acetylation.

## Results and discussion
### Overall architecture of NuA4

For our structural studies, we purified NuA4 from *S. cerevisiae* harboring a DNA fragment encoding a TAP tag at the 3′-end of the Esa1 acetyl transferase gene. The isolated NuA4 contained all 13 subunits of the complex (*Figure 1A*), as confirmed by sodium dodecyl sulfate–polyacrylamide gel electrophoresis (SDS–PAGE), mass spectrometry, and mass photometry (*Figure 1—figure supplement 1*, *Figure 1—figure supplement 2*). Despite reports that the HAT module of NuA4 can also form an independent complex in yeast (*Boudreault et al., 2003*), smaller particles that could correspond to that module alone were not clearly apparent in our cryo-EM images and analysis (*Figure 1*, *Figure 2* and *Figure 3*). Mass photometry data were not informative concerning the presence of the HAT module alone, as the detergent in our sample created a large peak in the low molecular weight (<300 kDa) region where the HAT module (195 kDa) would be expected (*Figure 1—figure supplement 1C*). Using single-particle cryo-EM image analysis we observe that the BS3 crosslinked NuA4 complex contains a rigid central hub with more flexible elements attached (*Figure 1B, C*). Focused refinement of the central hub allowed us to generate a density map for this region with an overall

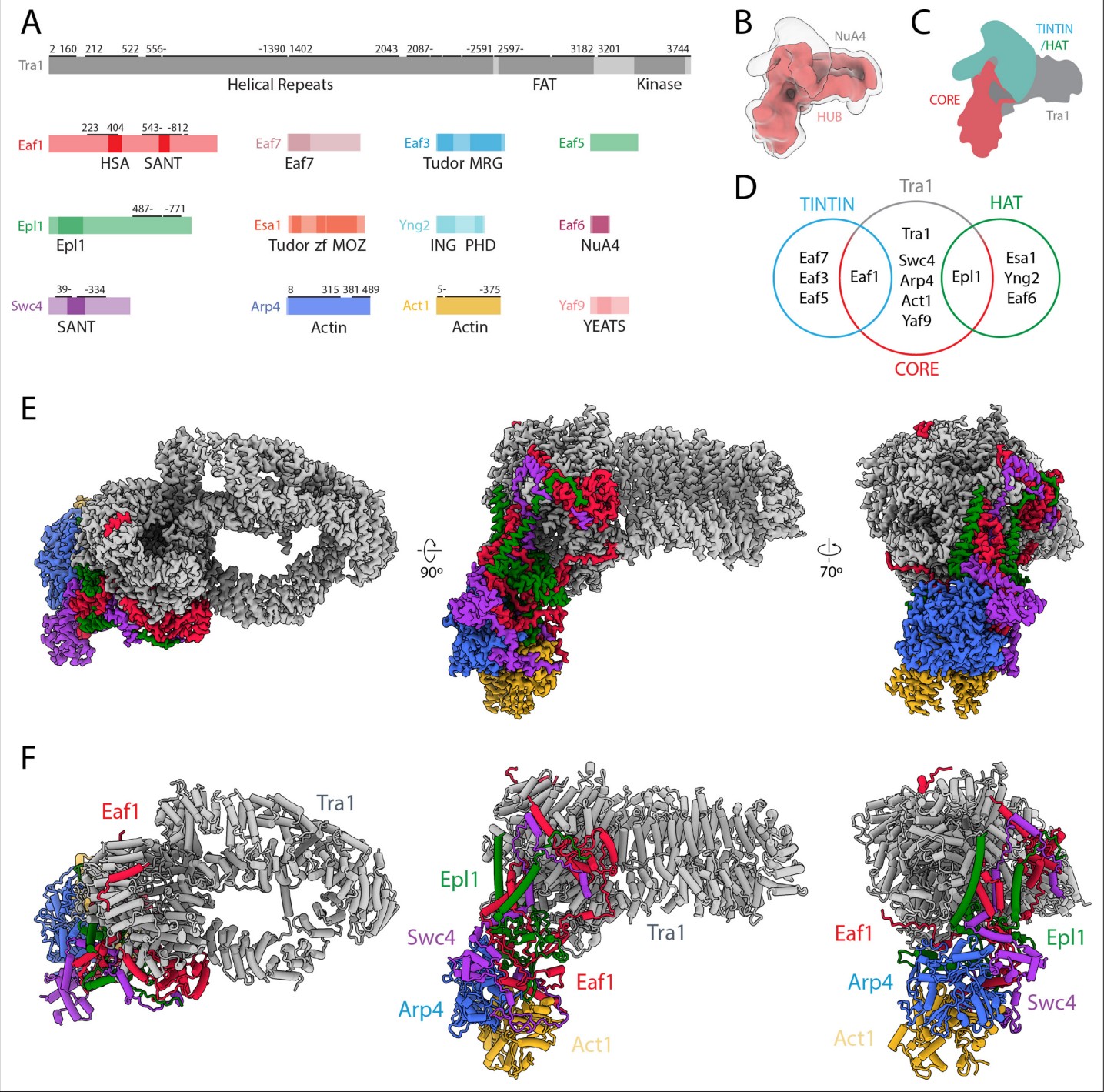

**Figure 1.** Structure of NuA4. (**A**) Domain map of NuA4 subunits. Modeled regions are marked with a black bar; numbers indicate starting and ending residues. (**B**) Cryo-electron microscopy (cryo-EM) map in red showing the best-defined parts of NuA4. A transparent lower-threshold cryo-EM map is overlaid to show the flexible density likely corresponding to the histone acetyltransferase (HAT)/Trimer Independent of NuA4 involved in Transcription Interactions with Nucleosomes (TINTIN) modules. (**C**) Cartoon representation of NuA4 modules. (**D**) Venn diagram showing NuA4 subunit organization across different complex modules. Subunits in the core attach to Tra1 and act to tether the TINTIN and HAT modules to the complex. (**E**) Cryo-EM map of the NuA4 hub with individual subunits colored. (**F**) Model of the NuA4 hub with individual subunits colored and labeled.

The online version of this article includes the following source data and figure supplement(s) for figure 1:

**Figure supplement 1.** Purification of NuA4.

**Figure supplement 1—source data 1.** Uncropped – sodium dodecyl sulfate–polyacrylamide gel electrophoresis (SDS–PAGE) (Bio-Rad 4–20%) of

*Figure 1 continued on next page*

*Figure 1 continued*

purified *S. cerevisiae* NuA4, stained with Flamingo (Bio-Rad).

**Figure supplement 2.** Protein domains of NuA4.

**Figure supplement 3.** Cryo-electron microscopy (cryo-EM) data collection and processing for NuA4.

**Figure supplement 4.** NuA4 structure model validation.

**Figure supplement 5.** Euler angle distribution and map anisotropy.

**Figure supplement 6.** Model fit for NuA4.

**Figure supplement 7.** Comparison with previous NuA4 model and CX-MS validation.

resolution of 3.1 Å (*Figure 1D, E*, *Figure 1—figure supplement 3*, *Figure 1—figure supplement 4*, *Figure 1—figure supplement 5*, *Figure 1—figure supplement 6*) and unambiguously identify the subunits Eaf1, Epl1, Swc4, Arp4, Act1, and the large Tra1 (*Figure 1F*, *Figure 1—figure supplement 6*). Missing from this more stable hub are most subunits of the HAT module, all subunits of the TINTIN module, and Yaf9, the three elements known to interact with nucleosomes (*Li et al., 2014*; *Shi et al., 2006*; *Xu et al., 2016*; *Chittuluru et al., 2011*; *Sathianathan et al., 2016*). Our cryo-EM map showed a large diffuse density above the FAT domain of Tra1 (*Figure 1B*) reflecting the presence of flexibly attached components of NuA4 that are likely to correspond to the missing HAT and TINTIN modules and the Yaf9 subunit.

Our cryo-EM-based structural model of the central hub was validated by mapping the previously reported chemical crosslinking and mass spectrometry (CX-MS) data of NuA4 (*Setiaputra et al., 2018*), which identified many crosslinks between subunits resolved in our structure of the central hub. Overlaying these crosslinks on our structure shows that all 82 identified interactions (52 intramolecular links and 30 intermolecular links) fall under the 30 Å cutoff for DSS crosslinks (*Figure 1—figure supplement 7*). This was not the case for the structure of NuA4 previously reported (*Wang et al., 2018*; *Figure 1—figure supplement 7*).

## Structure of the NuA4 central hub

The structure of the central hub of NuA4 includes Tra1 and a core of additional subunits that interact extensively with each other and tether all the rest of the components (*Figures 1 and 2*). The Tra1 subunit makes up most of the hub density (gray in *Figure 1F*), and has a very similar structure to that previously described (*Diaz-Santín et al., 2017*). It contains a large HEAT repeat (pink in *Figure 2A*), followed by the FAT (yellow in *Figure 2*) and pseudo-kinase domains (jointly also referred to as FATKIN) (cyan in *Figure 2A*). Eaf1, Epl1, and Swc4 within the core (red, green, and purple, respectively, *Figures 1 and 2A*) interact with Tra1 near the FATKIN region. Of these, Eaf1 is the primary Tra1 interaction partner, contributing 5500 Å² of the total 7700 Å² buried surface area between the core and Tra1 (*Figure 2B, C*). Arp4 and Act1 (orange and blue, respectively, *Figures 1 and 2A*) are the only components of the core that do not contact Tra1 (*Figure 2C*).

Structurally, the NuA4 core made of Eaf1, Epl1, Swc4, Arp4, and Act1, can be seen as containing a beta-cluster, a helical bundle, an Arp module, and two Eaf1 extensions (*Figure 2A*). The beta cluster is composed of β-strands from Eaf1 and Epl1 (*Figure 2A*, oval 1) and sits at the center of the core, surrounded by the helical bundle, Arp module and the Tra1 FAT domain. The helical bundle contains helices from Eaf1, Epl1, and Swc4 (*Figure 2A*, oval 2) and buttresses the interface of the Tra1 FAT and pseudo-kinase domains. The largest part of the core is the ARP module, which is made up of Act1, Arp4, the HSA helix of Eaf1, and Swc4 (*Figure 2A*, oval 3). Act1 and Arp4 bind to each other, end to end, to form a tight dimer. The Swc4 ring, which includes the SANT domain and a long-extended coil, wraps around this Act1/Arp4 dimer, while the long HSA (helicase-SANT-associated) helix of Eaf1 binds across the Arp4/Act1 dimer (similar to other HSA–Arp–actin interactions) (*Schubert et al., 2013*). Finally, two sets of Eaf1 extensions emanate from the helical bundle and the beta cluster. The Eaf1 extension from the helical bundle (*Figure 2A*, oval 4) binds the FAT domain of Tra1 using its SANT domain and two sets of latch helices. The Eaf1 extension from the beta cluster (*Figure 2A*, oval 5) binds the pseudo-kinase domain of Tra1 through an extended coil structure.

While most of the components that make up the central hub of NuA4 interact with one another, Eaf1 stands out as the primary scaffolding factor, as it has the largest surface interface with Tra1, Swc4, and Epl1 (*Figure 2B, C*). It also contributes the conserved HSA helix to the Arp module. The role of

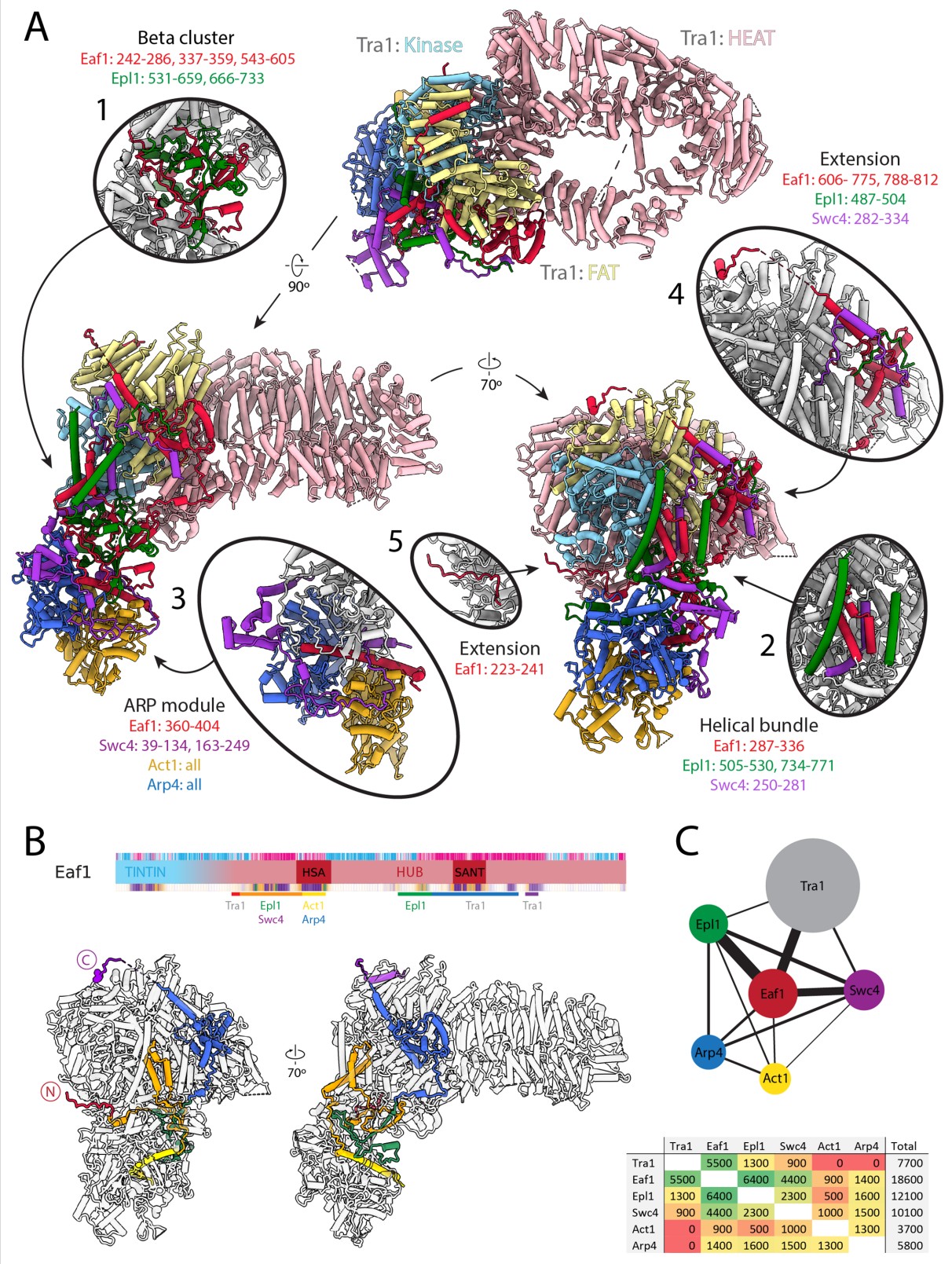

**Figure 2.** Architecture of the NuA4 central hub. (**A**) Structure of NuA4 with subunits of the core and Tra1 domains in different colors. Within Tra1, the pseudo-kinase domain is colored light blue, the FAT domain is colored pale yellow, and the HEAT domain is colored pink. The FAT and pseudo-kinase domains form the bulk of the interactions with the NuA4 core, highlighted in subpanels (1, 2, 4, and 5). Organization of the NuA4 core can be seen in subpanels (1, 2, and 3). Arp module containing Arp4 and Act1 assemble onto the HSA helix of Eaf1 (seen in subpanel 3). (**B**) Top: Eaf1 domain map, as

*Figure 2 continued on next page*

*Figure 2 continued*

introduced in *Figure 1—figure supplement 2*. Bars underneath are colored to indicate its protein interactions. Bottom: depiction of Eaf1 interaction with NuA4 subunits. Sections of Eaf1 are colored in a rainbow from N to C terminus with different colors representing regions with different protein–protein interactions. (**C**) Top: schematic representation of contacts between NuA4 subunits. The width of each line is proportional to the contact area between subunits. Bottom: table showing the contact area (Å²) between NuA4 subunits, colored from red (minimal contact) to green (maximal contact).

Eaf1 as a scaffolding subunit within NuA4 is in good agreement with previous genetic and biochemical studies that show that its deletion results in loss of NuA4 complex assembly (*Auger et al., 2008*; *Mitchell et al., 2008*).

## The central hub tethers the nucleosome-interacting components of NuA4

Our cryo-EM reconstruction resolved the structure of the rigid hub of NuA4 containing the core and Tra1, but the three chromatin-interacting components – the HAT and TINTIN modules and the Yaf9 subunit (*Steunou et al., 2016*; *Klein et al., 2018*) – are missing. These missing parts are flexibly tethered to the core through Epl1, Eaf1, and Swc4, respectively. The missing components likely make up the diffuse density observed above the Tra1 FATKIN domain in our cryo-EM map, as all three have been shown to crosslink with the FATKIN domain (*Figure 3A*).

Reconstitution experiments and the crystal structure of the HAT module show that it is composed of Yng2, Eaf6, and Esa1, as well as residues 121–400 of Epl1 (*Xu et al., 2016*). This segment of Epl1 is connected to the rest of the protein in the core of NuA4 via a poorly conserved and predicted unstructured region of about 90 amino acids (residues 400–487) (*Figure 3B*), in agreement with the lack of a fixed position for the HAT module with respect to the core. The N-terminus of the region of Epl1 integrated into the core is located near the FAT domain of Tra1, making the HAT module a likely candidate for the diffuse density we see in our structure above the FAT domain (*Figure 3A*).

There is little structural information concerning the yeast TINTIN module, which is composed of Eaf3, Eaf5, and Eaf7 and tethered to the core of the complex through Eaf1 (*Setiaputra et al., 2018*). We carried out AlphaFold2 prediction (*Jumper et al., 2021*; *Mirdita et al., 2022*) of the NuA4 TINTIN module. The resulting structure shows the module is split into three main ordered segments: the Eaf3 chromodomain, the Eaf3 MRG/Eaf5 C-term/Eaf7, and the Eaf5 N-term/Eaf1 (*Figure 3A*). These three parts appear interconnected through flexible linkers, making the TINTIN module highly extended. This feature is also captured in the CX-MS data (*Setiaputra et al., 2018*), which shows crosslinks within each of these three parts but no crosslinks between them (*Figure 3—figure supplement 1*). The main segment of the TINTIN module, composed of the Eaf3 MRG domain, Eaf7, and the C-terminus of Eaf5 (aa 179–279) holds the three components of the module together. The interactions between the Eaf3 MRG and Eaf7 are structurally similar to those in the human homolog (*Xie et al., 2015*). The TINTIN module is connected to the NuA4 core through the interaction of the N-termini of Eaf5 (aa 1–139) and Eaf1 (aa 28–91). The region of Eaf1 that interacts with the TINTIN module is separated from the Eaf1 region located within the core of NuA4 by a segment of approximately 130 amino acids that is predicted to be unstructured (*Figure 3B*). The most N-terminal part of Eaf1 integrated within the core of NuA4 is located near the back of Tra1, near the pseudo-kinase domain, far from the diffuse density seen above the FATKIN domain (*Figure 3A*). However, the long linker length between the integrated region of Eaf1 and the region that is predicted to interact with the TINTIN module could easily span the distance to the diffuse density and enable the observed crosslinks between Eaf5 and Tra1 to form. So, in addition to the HAT domain, the TINTIN domain could also make up part of the diffuse density seen above the FAT domain in our cryo-EM structure (*Figure 3A*).

Finally, CX-MS data show that Yaf9 interacts with the C-terminus of Swc4 (residues 359–476) (*Setiaputra et al., 2018*). Previous binding studies have shown that the YEATs domain of Yaf9 is capable of binding acetylated lysine residues, with the highest affinity for H3K27ac (*Klein et al., 2018*). Such interaction has been visualized in an X-ray crystal structure of the Yaf9 YEATS domain bound to an acetylated peptide (*Klein et al., 2018*). The AlphaFold2 prediction of Yaf9 and the C-terminus of Swc4 shows that the Swc4 binds the Yaf9 β-sandwich on the opposite side of the histone-binding face and forms a coil–coil structure with a protruding C-terminal helix on Yaf9 (*Figure 3A*, *Figure 3—figure supplement 1*). The region of Swc4 that is integrated into the core of NuA4 and interacts with Yaf9 is linked via a poorly conserved and predicted unstructured region of about 20 amino acids (*Figure 3B*).

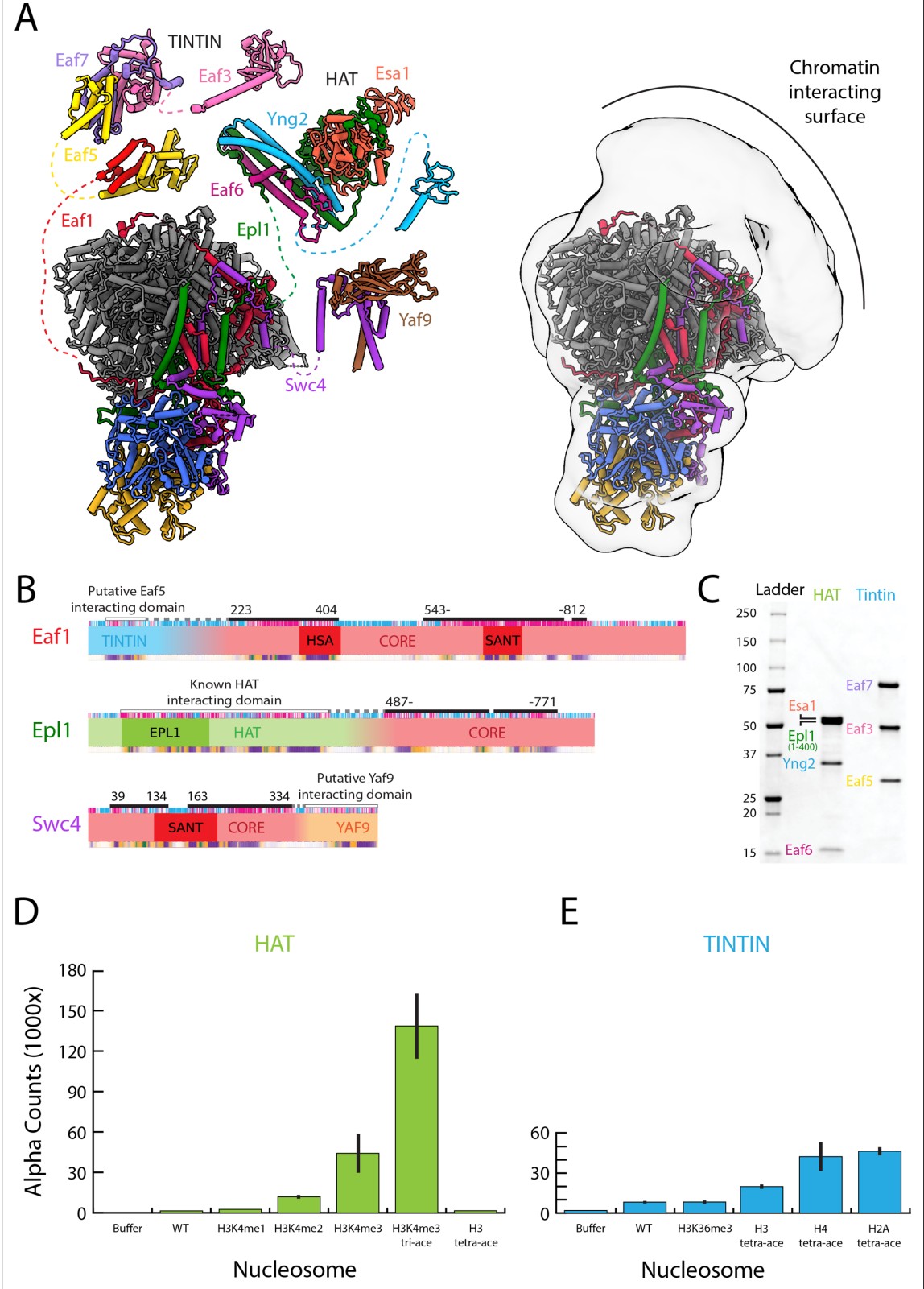

**Figure 3.** The NuA4 central hub tethers the nucleosome-interacting modules. (**A**) Model for NuA4 complex organization (NuA4 central hub model from the present cryo-electron microscopy (cryo-EM) structure, with the rest of the models from AlphaFold2 prediction) (*Jumper et al., 2021*; *Mirdita et al., 2022*). Spatial constraints are imposed on the position of flexible modules by the length of the linker to the connecting amino acids resolved in the structure. Additional low-resolution density adjacent to the NuA4 hub suggests the approximate location of the flexible modules. (**B**) Domain

*Figure 3 continued on next page*

*Figure 3 continued*

map showcasing the subunits that link the Trimer Independent of NuA4 involved in Transcription Interactions with Nucleosomes (TINTIN), histone acetyltransferase (HAT), and YAF9 modules to the HUB. (**C**) Sodium dodecyl sulfate–polyacrylamide gel electrophoresis (SDS–PAGE) (Bio-Rad 4–20%) of purified NuA4 TINTIN and HAT modules, stained with InstantBlue (Expedeon). (**D**) dCypher assay results of nucleosome discovery screen for the purified HAT module. Error bars calculate from duplicate experiments, (**E**) dCypher assay result of nucleosome discovery screen for the purified TINTIN module. Error bars calculate from duplicate experiments.

The online version of this article includes the following source data and figure supplement(s) for figure 3:

**Source data 1.** Uncropped – sodium dodecyl sulfate–polyacrylamide gel electrophoresis (SDS–PAGE) (Bio-Rad 4–20%) of purified NuA4 Trimer Independent of NuA4 involved in Transcription Interactions with Nucleosomes (TINTIN) and histone acetyltransferase (HAT) modules, stained with InstantBlue (Expedeon).

**Figure supplement 1.** AlphaFold2 prediction and validation of the flexible NuA4 modules.

**Figure supplement 2.** NuA4 and SAGA both have flexible nucleosome engaging domains.

**Figure supplement 3.** NuA4-negative stain data processing.

**Figure supplement 4.** SAGA-negative stain data processing.

**Figure supplement 5.** Complete results of dCypher nucleosome discovery screen.

So, like the HAT and TINTIN module the Yaf9 could also occupy the diffuse density above the FAT domain of Tra1.

NuA4 is not the only HAT to feature a modular structure composed of a central core tethering various flexibly chromatin-interacting and modifying modules. SAGA, the other major multisubunit HAT in yeast, has been reported to have similar flexible domains (*Setiaputra et al., 2015*). To directly compare the relative flexibility of the two complexes we performed negative stain electron microscopy of BS3 crosslinked HATs, NuA4, and SAGA (*Figure 3—figure supplement 2*, *Figure 3—figure supplement 3*, *Figure 3—figure supplement 4*). The negative stain structures show strong density for the central hub of the two complexes, both of which contain Tra1 and a core that tethers the more flexible modules. The negative strain structures show that the flexible regions of NuA4 appear to be more diffuse (lower occupancy) than the HAT and DUB modules of SAGA, and are more similar to what is seen as corresponding to the Spt8 subunit (*Figure 3—figure supplement 2A, C*; *Setiaputra et al., 2015*). Due to the similarities between NuA4 and SAGA, both containing the activator targeting Tra1 subunit and having a similar overall architecture that includes flexible attachment of chromatin-binding and modifying modules, we propose that these two HAT complexes are likely to have a similar chromatin targeting mechanism.

## The HAT and TINTIN modules recognize active transcriptional marks

Due to the small amounts of NuA4 that can be purified from yeast and the generally weak chromatin-binding activity of the complex, testing the chromatin recognition capabilities of NuA4 is difficult (*Li et al., 2007*). To overcome these constraints, we reconstituted the HAT and TINTIN modules separately through recombinant expression and utilized the dCypher approach (*Weinberg et al., 2019*; *Marunde et al., 2022b*; *Marunde et al., 2022a*) to interrogate the binding of the two modules against PTM-defined histone peptides (data not shown) and nucleosomes (*Figure 3C*).

The HAT module contains three putative chromatin interacting domains: a chromodomain and the HAT domain in Esa1 (the later combines a zinc-finger and MOZ-type HAT domain), and a PHD in Yng2. Previous reports have shown that the PHD domain of Yng2 can bind H3K4me3, while the HAT domain binds the histone octamer surface of the nucleosome (*Xu et al., 2016*; *Steunou et al., 2016*). Although we did see increasing HAT module affinity for nucleosomes with a greater number of methyl groups on H3K4, affinity for nucleosomes containing both H3K4me3 and acetylation marks on H3 (K9, K14, and K18) demonstrated ~3× binding preference over H3K4me3 alone (*Figure 3D*, *Figure 3*, *Figure 4* and *Figure 5A*). Tetra-acetylated H3 nucleosomes (K4, K9, K14, and K18) as well as singly acetylated K9, K14, and K18 nucleosomes were not bound by the HAT module, indicating that acetylation on its own is poorly or not recognized (*Figure 3D*). Instead, the acetylated lysines are likely to reduce overall interactions between the histone tails and DNA, allowing the tails greater accessibility by the HAT module of the H3K4me3 mark (*Morrison et al., 2018*; *Marunde et al., 2022a*; *Morgan et al., 2021*; *Jain et al., 2022*).

Within the TINTIN module, the only predicted chromatin interacting domain is the Eaf3 chromo-domain. Eaf3 is also present in the yeast HDAC RPD3S, where it has been proposed to recognize H3K36me3 modified nucleosomes (*Huh et al., 2012*; *Lee et al., 2013*; *Ruan et al., 2015*; *Keogh et al., 2005*). However, we found that TINTIN has overall weak affinity for nucleosomes and no specificity for H3K36me3, an observations made by others for NuA4 as a whole (*Figure 3E*, *Figure 3—figure supplement 5B*; *Li et al., 2007*; *Xu et al., 2008*). The only slight preference of the complex appears to be for H2A and H4 acetylated nucleosomes, though with the caveat that this was observed under conditions of high binding background (*Figure 3E*, *Figure 3—figure supplement 5B*). Interestingly, these two sets of histone modifications are made by the HAT module of NuA4.

Our binding studies show that NuA4 prefers H3K4me3-containing nucleosomes that are hyper-acetylated on H3. Acetylation of H3 in yeast is largely conferred by SAGA, both in the context of transcription and double strand break repair (*Grant et al., 1997*; *Kuo et al., 1998*). Thus, our binding studies suggest that NuA4 activity likely follow that of SAGA, while it would precede the recruitment of factors like TFIID or SWR1 that contain reader modules for H4 acetylation (*Matangkasombut and Buratowski, 2003*; *Ladurner et al., 2003*; *Durant and Pugh, 2007*).

## Structural comparison of NuA4 with other complexes

Many of the components of yeast NuA4 are shared with other complexes (*Figure 4A*; *Doyon and Côté, 2004*; *Grant et al., 1998*). Part of the ARP module is also found in the SWR1 and INO80 chromatin remodelers (*Mizuguchi et al., 2004*; *Shen et al., 2000*), while the large activator targeting Tra1 subunit is shared with SAGA, the other major HAT with functions in transcription (*Grant et al., 1998*). The HAT and TINTIN can also exist outside of NuA4 (*Boudreault et al., 2003*; *Rossetto et al., 2014*). Of note, the higher eukaryotic TIP60 complex combines components from the yeast NuA4 and SWR1 complexes (*Auger et al., 2008*).

The yeast INO80 and SWR1 complexes function as histone remodelers/histone exchangers during transcription initiation and DNA repair (*Mizuguchi et al., 2004*; *Shen et al., 2000*; *Morrison and Shen, 2009*; *van Attikum et al., 2004*). These functions are in part facilitated by NuA4, which acetylates the nucleosomes that are to be remodeled or have their histones exchanged (*Ranjan et al., 2013*; *Altaf et al., 2010*; *Krogan et al., 2004*; *Downs et al., 2004*). In SWR1, the ARP module contains Act1, Arp4, Yaf9, and the HSA helix of Swr1 replacing Eaf1 (*Figure 4B*; *Wu et al., 2009*; *Cao et al., 2016*). In INO80 the ARP module contains Act1, Arp4, and the HSA helix of Ino80 (*Figure 4B*; *Knoll et al., 2018*). In both SWR1 and INO80, the ARP module is flexible with respect to the core of the complex, with the HSA helix of Swr1 and Ino80 predicted to be solvent exposed (*Knoll et al., 2018*; *Brahma et al., 2018*; *Willhoft et al., 2018*; *Ayala et al., 2018*; *Eustermann et al., 2018*). From biochemical studies, the solvent exposed side of the HSA helix in INO80 seems to bind extranucleosomal DNA, as also predicted for SWR1 (*Knoll et al., 2018*; *Brahma et al., 2018*). In contrast, in NuA4, the ARP module surface with the HSA helix is buried in the core of the complex, where it appears to serve largely a structural role (*Figure 4B*).

In *S. cerevisiae* the multisubunit NuA4 and SAGA acetyltransferase complexes both include the Tra1 transactivation-binding protein (*Allard et al., 1999*; *Grant et al., 1998*), while they do not share any other subunits and acetylate different histone tails (NuA4 targeting H2A/H4; SAGA targeting H3). Despite their compositional and functional differences, both complexes contain a central core that interacts with Tra1, primarily through the FAT domain, and anchors the rest of the functional domains. We describe here how the NuA4 core contacts Tra1 through an extensive interface involving Eaf1, Swc4, and Epl1 (*Figure 4C*). In contrast, yeast SAGA has a much smaller Tra1 interface involving Spt20 and Taf12 (*Figure 4D*; *Papai et al., 2020*; *Wang et al., 2020*). The larger interface between Tra1 and the NuA4 core results in a more rigid connection between these two modules compared with yeast SAGA (*Figure 4C, D*; *Papai et al., 2020*; *Wang et al., 2020*).

The human TIP60 complex encompasses the functionalities of yeast NuA4 and SWR1 complexes (*Doyon and Côté, 2004*; *Auger et al., 2008*), with the metazoan EP400 being the key protein responsible for merging the two complexes present in lower eukaryotes (*Auger et al., 2008*, *Figure 4—figure supplement 1A, B*). Based on the domain organization of EP400 and the structures of SWR1 and NuA4, we propose a putative organization of the TIP60 complex (*Figure 4—figure supplement 1C*, *Willhoft et al., 2018*). The N-terminus of EP400 would be within the NuA4 portion of the TIP60 complex, forming part of the core and contributing the HSA helix to the ARP module. The next

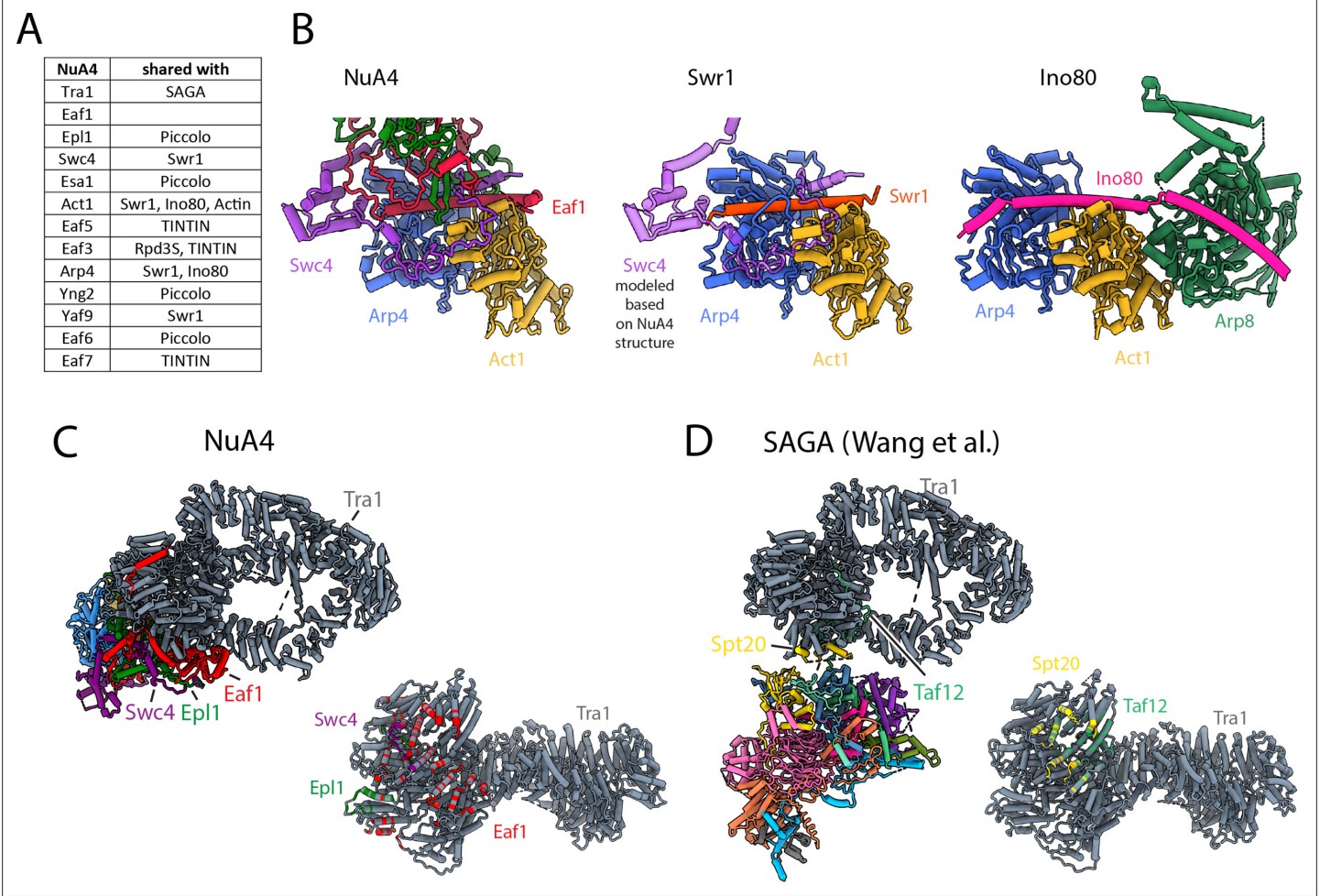

**Figure 4.** NuA4 subunits shared with other *Saccharomyces cerevisiae* complexes. (**A**) Table showing protein subunits of NuA4 and the complexes in yeast that share these subunits. (**B**) Comparison of Arp4–Act1 interactions with HSA helix in NuA4 (Eaf1 subunit), SWR1 (Swr1 subunit), and INO80 (Ino80 subunit) (*Cao et al., 2016*; *Knoll et al., 2018*). (**C**) Left/top: model of NuA4 depicting attachment of the core to Tra1. Right/bottom: the Tra1 subunit of NuA4 colored according to which protein chain contacts it. (**D**) Left/top: model of SAGA depicting attachment of the core to Tra1. Right: the Tra1 subunit in SAGA colored according to which protein chain contacts it (*Papai et al., 2020*; *Wang et al., 2020*).

The online version of this article includes the following figure supplement(s) for figure 4:

**Figure supplement 1.** A model of *H.s.*TIP60.

section of EP400 would go into the SWR1 portion of the complex, where it would contribute the Snf2 domain of the ATPase module, the RUVBL interaction domain, and the helicase domain of the ATPase module. From there, the C-terminal segment of EP400 would return to the NuA4 portion of the complex, where it would make further contributions to the core, ending with the interaction of its SANT domain with the TRRAP subunit (Tra1 homolog).

## Model of nucleosome selection and acetylation by NuA4

From our structural and biochemical data, we propose a sequential model of NuA4 recruitment to target nucleosomes (*Figure 5*). The first step would be the binding of site-specific transcription factors, capable of interaction with the Tra1 subunit to recruit NuA4. The NuA4 flexible HAT and reader modules (TINTIN and Yaf9) would then be able to select neighboring nucleosomes containing preferred modifications, such as H3K4me3 with acetyl groups on H3 K9, K14, and K18. Once a nucleosome is selected by the complex, the HAT module would acetylate histones H2A and H4.

This NuA4 mechanism of recruitment is likely similar for the human TIP60 homolog and may also be shared by other HAT complexes such as SAGA and CBP/p300. Like NuA4, SAGA, and CBP/p300 also have their activator-binding and HAT components flexibly separated (*Papai et al., 2020*; *Wang et al.,*

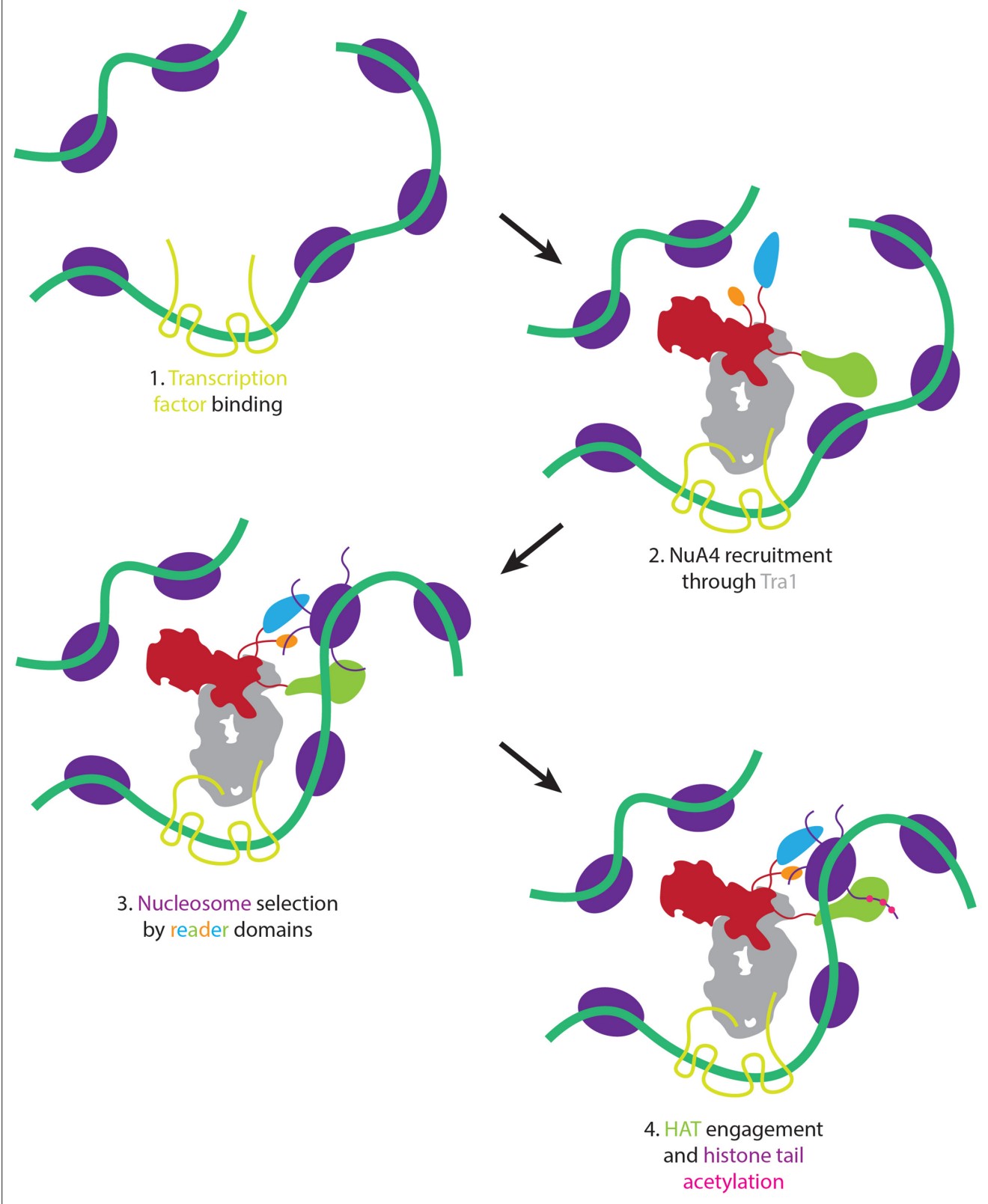

**Figure 5.** Proposed model of nucleosome selection and propagation of acetylation. Model of NuA4 chromatin localization and histone acetylation. NuA4 is recruited to genomic loci through interaction of Tra1 with site-specific transcription factors. Once recruited, the flexible reader domains interrogate nearby nucleosomes. Nucleosomes bearing the proper chemical marks are preferentially acetylated.

*2020*; *Fuxreiter et al., 2008*; *Dyson and Wright, 2016*). This structural arrangement likely means that recruitment of the complexes does not constrain the HAT module to act on a specific nucleosome, such as the most adjacent one, but could allow for the complex to target specifically modified nucleosomes within the neighboring chromatin environment.

## Methods
### Protein purification
#### NuA4 and SAGA

NuA4 and SAGA were purified from *S. cerevisiae* using a modified TAP purification as described (*Puig et al., 2001*). A strain modified with a TAP tag on the C terminus of ESA1 or SPT7 (*GE Dharmacon*) was grown at 30°C in YPD. 12 l of cells were harvested at OD ~6.0 and lysed using a Planetary Ball Mill (Retsch PM100). The ground cells were resuspended in lysis buffer (100 mM HEPES [4-(2-hydroxyethyl)-1-piperazineethanesulfonic acid] pH 7.9, 2 mM $MgCl_2$, 300 mM KCl, 10 µM leupeptin, 0.5 mM PMSF [phenylmethylsulfonyl fluoride], 1× EDTA-free complete protease inhibitor [ethylenediaminetetraacetic acid], 0.05% NP-40, 10% glycerol) and dounced until homogenous. The lysate was centrifuged for 1 hr at 15,000 × *g*, and clarified lysate passed through a clean column frit, and aliquoted into 50 mL Falcon tubes. To each 50 ml tube of lysate, 200 µl of washed IgG resin was added and allowed to incubate overnight at 4°C with gentle rocking. After incubation, resin was collected by centrifuging tubes at 2000 × *g* for 10 min and removing the supernatant. Resin was resuspended in 400 µl of TAP buffer (20 mM HEPES pH 7.9, 2 mM $MgCl_2$, 300 mM KCl, 10 µM leupeptin, 0.5 mM PMSF, 0.05% NP-40, 10% glycerol) and transferred to a microcentrifuge tube. Tubes were centrifuged for 2 min at 2000 × *g* and supernatant removed. Resin was washed 5× with TAP buffer. To washed resin, one volume of TAP buffer was added, then TCEP (tris(2-carboxyethyl)phosphine) was added to a concentration of 1 mM. 40 µg of TEV protease (Macrolab) was added for every 500 µl of resin, and tubes were let incubate at 4°C overnight.

Tubes were centrifuged for 2 min at 2000 × *g*, and the supernatant transferred to another tube as the first elution. Resin was washed 2× with one volume of TAP buffer, and supernatant saved as second and third elutions. All elutions were run on SDS–PAGE and stained using Flamingo protein stain (Bio-Rad). Those containing NuA4 complex (based on SDS–PAGE and later confirmed by mass spectrometry) were pooled, and the calcium concentration increased by adding calcium chloride to 2 mM final by fast dilution. Pooled elutions were incubated with 150 µl washed CBP resin at 4°C for 4 hr. Tubes were centrifuged for 2 min at 2000 × *g*, and the supernatant removed. Resin was washed 5× with one volume of CBP wash buffer (20 mM HEPES pH 7.9, 2 mM $MgCl_2$, 300 mM KCl, 2 mM $CaCl_2$, 10 µM leupeptin, 0.01% NP-40, 10% glycerol). To washed resin, one volume of CBP elution buffer (20 mM HEPES pH 7.9, 2 mM $MgCl_2$, 300 mM KCl, 2 mM EGTA (ethylene glycol-bis(β-aminoethyl ether)-N,N,N',N'-tetraacetic acid), 10 µM leupeptin, 0.01% NP-40, 10% glycerol) was added and let incubate for 30 min. Tubes were centrifuged for 2 min at 2000 × *g*, and the supernatant was transferred to another tube. Elution was repeated three times. Elutions were run on SDS–PAGE and Flamingo stained, with those containing NuA4 or SAGA flash-frozen in liquid nitrogen.

#### HAT

The NuA4 HAT module was purified from Hi5 insect cells (*Trichoplusia ni*) coexpressing Yng2, Eaf6, Esa1, and Epl1 (1–400). All genes were codon optimized and synthesized as gBlocks (*IDT*), with Epl1 harboring a C-terminal HIS tag. Synthesized genes for Yng2, Eaf6, and Esa1 were cloned into plasmid 438 A. while Epl1 (1–400) was cloned into plasmid 38Q (N-terminal MBP – TEV site) (*Gradia et al., 2017*). Genes were then combined into a single plasmid (*Gradia et al., 2017*; *Irwin et al., 2012*) used to generate bacmids by transformation into DH10MultiBac cells (Macrolab). Purified bacmids were transfected into Sf9 cells (*Spodoptera frugoperda*) using FuGene (Promega) and baculoviruses amplified for two rounds before protein expression in Hi5 cells.

Harvested cells were resuspended in 50 ml of lysis buffer (25 mM HEPES 7.9, 150 mM NaCl, 2.5 mM $MgCl_2$, 10% glycerol, 1× cOmplete EDTA-free protease inhibitor [Roche]) and sonicated for a total processing time of 5 min at 40% power. 5 µl of benzonase (Sigma) was added and incubated at 4°C for 20 min with gentle rocking. NaCl was added to a final concentration of 300 mM, as well as 50 µl of 10% Triton-X and 50 µl of NP-40 substitute. Lysate was incubated at 4°C for 10 min with gentle

rocking. Lysate was centrifuged for 50 min at 18,000 rpm and supernatant was removed. Imidazole, to a final concentration of 10 mM was added to supernatant, which was then incubated with 1.5 ml washed and packed nickel resin (GoldBio) at 4°C for 1 hr with gentle rocking. Beads were removed and washed five times with 10 ml of wash buffer (25 mM HEPES 7.9, 300 mM NaCl, 2.5 mM MgCl$_2$, 10% glycerol, 25 mM imidazole). Protein was eluted with elution buffer (25 mM HEPES 7.9, 300 mM NaCl, 2.5 mM MgCl$_2$, 10% glycerol, 250 mM imidazole). Elutions were analyzed with SDS–PAGE, and samples containing NuA4 HAT were pooled. TCEP was added to a concentration of 1 mM, and pooled elutions were incubated with 4 ml of washed and packed amylose resin (GE) at 4°C for 1 hr with gentle rocking. Beads were removed and washed five times with 10 ml of amylose wash buffer (25 mM HEPES 7.9, 300 mM NaCl, 2.5 mM MgCl$_2$, 10% glycerol). Protein was eluted with amylose elution buffer (25 mM HEPES 7.9, 100 mM NaCl, 2.5 mM MgCl$_2$, 10% glycerol, 20 mM maltose). Fractions containing NuA4 HAT were pooled and concentrated to 9 mg/ml. Sample was incubated with TEV protease (Macrolab) overnight at 4°C without any motion. Digested sample was centrifuged at max speed for 10 min, then 500 µl was loaded onto a 24 ml Superdex 200 increase size exclusion column equilibrated into sizing buffer (25 mM HEPES, 2.5 mM MgCl$_2$, 100 mM NaCl, 10% glycerol). Peaks were analyzed with SDS–PAGE and samples containing the intact NuA4 HAT complex were pooled, concentrated to a final concentration of 5 mg/ml and frozen.

## TINTIN

The NuA4 TINTIN module was purified from BL21STAR (Macrolab) *E. coli* cells coexpressing Eaf3, Eaf5, and Eaf7 in a polycistronic construct. The polycistronic block was synthesized in two gBlock (IDT) with the individual genes codon optimized and Epl7 harboring a C-terminal FLAG tag. The two gBlocks were cloned together into the 2G-T plasmid (Addgene #29707), so that Eaf3 would have a N-terminal HIS-GST-TEV site (*Irwin et al., 2012*). The constructed plasmids were transformed into BL21STAR cells. Cells were grown in TB media at 37°C till they reached an OD of 0.8, at which point the cells were cooled to 18°C and protein expression was induced with 0.1 mM IPTG (isopropyl β-d-1-thiogalactopyranoside). Cells were harvested after 16 hr.

Cells were resuspended in 50 ml of lysis buffer (25 mM HEPES 7.9, 150 mM NaCl, 2.5 mM MgCl$_2$, 10% glycerol, 1× cOmplete EDTA-free protease inhibitor) and sonicated for a total processing time of 5 min at 40% power. 5 µl of benzonase was added and incubated at 4°C for 20 min with gentle rocking. NaCl was added to a final concentration of 300 mM, as well as 50 µl of 10% Triton-X and 50 µl of NP-40 substitute. Lysate was incubated at 4°C for 10 min with gentle rocking. Lysate was centrifuged for 50 min at 18,000 rpm and supernatant was removed. Imidazole, to a final concentration of 10 mM was added to supernatant, which was then incubated with 1.5 ml washed and packed nickel resin (GoldBio) at 4°C for 1 hr with gentle rocking. Beads were removed and washed five times with 10 mL of wash buffer (25 mM HEPES 7.9, 300 mM NaCl, 2.5 mM MgCl$_2$, 10% glycerol, 25 mM imidazole). Protein was eluted with elution buffer (25 mM HEPES 7.9, 100 mM NaCl, 2.5 mM MgCl$_2$, 10% glycerol, 250 mM imidazole). Elutions were analyzed with SDS–PAGE, and samples containing TINTIN were pooled. Sample was loaded onto a 1 ml HiTrap Q column and eluted via salt gradient. Fractions containing TINTIN were pooled and loaded onto a 1 ml Heparin column and eluted via salt gradient. Fractions containing TINTIN were pooled, concentrated, and loaded onto a 24 ml Superdex 200 increase size-exclusion column. Peaks were analyzed with SDS–PAGE and samples containing the intact TINTIN complex were pooled, concentrated to a final concentration of 2.5 mg/ml and frozen.

## Mass photometry

Mass photometry experiment were performed on a Refeyn Two$^{MP}$ mass photometer (Refeyn Ltd, Oxford, UK; *Sonn-Segev et al., 2020*). Data were collected using Refeyn Acquire$^{MP}$ (v.2022 R1) software in the large image size mode. Mass calibration was performed using bovine serum albumin (BSA, Thermo Fisher cat. #23209) and urease (Sigma-Aldrich cat. #T9145) standards with 66, 132, 272, and 544 kDa as mass references in Discover$^{MP}$ (v. 2022 R1) software. Mass calibration was performed by focusing instrument with 15 µl of 1× phosphate-buffered saline using droplet dilution method and adding 5 µl of premixed BSA (0.1 µM) and urease (0.1 mg/ml) standards for the measurement.

To measure the mass of Nu4 complex we have performed a series of measurements for sample buffer (20 mM HEPES pH 7.9, 150 mM KCl, 2 mM MgCl$_2$, and 10% glycerol), sample buffer with NP-40 (0.01% NP-40), and Nu4 complex resuspended in sample buffer with NP-40. The instrument was

focused using the droplet dilution method, with 15 µl of sample buffer followed by collection of a 60-s movie. Then the instrument was refocused, 5 µl of buffer were pipetted off and replaced with 10 µl of ~100 nM Nu4 complex, followed by collection of a 60-s movie.

To determine the effect of the detergent present in the Nu4 sample, the same procedure was applied but instead of Nu4 complex, 10 µl of sample buffer with NP-40 were added to the sample buffer.

Particle landing events were analyzed in Discover$^{MP}$ with default parameters. Data are presented as histograms with Gaussian fit (*Figure 1—figure supplement 1C*). Displayed masses correspond to the single mass value present at the highest point of the fitted curve.

## Negative stain sample preparation, data collection, and data processing

Purified NuA4 complex was diluted 1:2 with negative stain crosslinking buffer (20 mM HEPES pH 7.9, 0.1 mM EDTA, 2 mM $MgCl_2$, 1% trehalose, 1% glycerol, 75 mM KCl, 1 mM TCEP, 0.01% glutaraldehyde) and allowed to crosslink on ice for 5 min. 4 µl was then applied to a glow-discharged continuous carbon grid for 5 min and stained using uranyl formate. The negative stain dataset was collected on a Tecnai F20 microscope (FEI) operated at 120 keV and equipped with an Ultrascan 4000 camera (Gatan). Data were collected using Leginon data acquisition software (*Suloway et al., 2005*). The contrast transfer function (CTF) parameters were estimated using Gctf (version 1.16) and particles were picked using Relion (*Zhang, 2016*; *Zivanov et al., 2018*). Data processing was done using Relion (version 3.1) (*Zivanov et al., 2018*). Extracted particles were subjected to 2D classification, ab initio model generation, and 3D classification to identify different conformation states of the complex and to visualize the flexibility of its submodules.

Purified SAGA complex was diluted 1:2 with negative stain crosslinking buffer (20 mM HEPES pH 7.9, 0.1 mM EDTA, 2 mM $MgCl_2$, 1% trehalose, 1% glycerol, 75 mM KCl, 1 mM TCEP, 0.01% glutaraldehyde) and allowed to crosslink on ice for 5 min. 4 µl was then applied to a glow-discharged continuous carbon grid for 5 min, stained using uranyl formate, and tilted negative stain dataset collected on a Tecnai F20 microscope (FEI) operated at 120 keV/equipped with an Ultrascan 4000 camera (Gatan). Data were collected using Leginon data acquisition software (*Suloway et al., 2005*). The CTF parameters were estimated using Gctf (version 1.16), particles picked using Relion (*Zhang, 2016*), and data processing done using Relion (version 3.1) (*Zivanov et al., 2018*). Extracted particles were subjected to ab initio model generation and 3D classified to identify different conformation states of the complex and visualize the flexibility of its submodules.

## Cryo-EM sample preparation

For cryo-EM sample preparation we used a Vitrobot Mark IV (FEI). NuA4 was crosslinked on ice using 1 mM BS3 (Thermo Fisher Scientific) for 15 min before 4 µl of sample was applied to a graphene oxide coated 1.2/1.3 UltrAuFoil grids (Quantifoil) at 4°C under 100% humidity (*Patel et al., 2021*). The sample was immediately blotted away using Whatman #1 for 2 s at 5 N force and then immediately plunge frozen in liquid ethane cooled by liquid nitrogen.

## Cryo-EM data collection

Grids were clipped and transferred to the autoloader of a Talos Arctica electron microscope (Thermo Fisher Scientific) operating at 200 keV acceleration voltage. Images were recorded with a K3 direct electron detector (Gatan) operating in superresolution mode at a calibrated magnification of 44,762 (.5585 Å/pixel), using the SerialEM data collection software (*Mastronarde, 2005*). 38-frame exposures were taken at 0.065 s/frame, using a dose rate of 12.98 e$^-$/pixel/s (1.05 e$^-$/Å$^2$/frame), corresponding to a total dose of 40 e$^-$/Å$^2$ per micrograph (*Figure 1—figure supplement 3*). A total of 9701 movie were collected from a single grid.

## Cryo-EM data processing

All data processing was performed using Relion3 (version 3.0) (*Zivanov et al., 2018*). Whole movie frames were aligned and binned by 2 (1.117 Å/pixel) with MotionCor2 to correct for specimen motion (*Zheng et al., 2017*). The CTF parameters were estimated using Gctf (*Zhang, 2016*). 3,276,565 particles were picked with Gautomatch (version 0.53, from K. Zhang, MRC-LMB, Cambridge). Particles

were extracted binned by 4 (4.468 Å/pixel) and subjected to two-dimensional classification to remove ice, empty picks and graphene oxide creases, which resulted in 2,296,668 particles. These particles were reextracted binned by 3 (3.351 Å/pixel) and subjected to three-dimensional classification, and the best class, containing 635,860 particles, was selected for further processing. The particles were reextracted, binned by 1.2 (1.3404 Å/pixel) and subjected to three-dimensional refinement, resulting in a 3.98-Å map. Particles were subjected to CTF refinement and particle polishing before (*Zivanov et al., 2018*; *Zivanov et al., 2019*) performing another 3D refinement, which resulted in reconstructions at 3.1 Å. The core was then subjected to multibody refinement by masking the complex into three parts (core, Tra1-FATKIN, and Tra1-HEAT), which refine to 2.9, 2.9, and 3.4 Å (*Nakane et al., 2018*). While the core and Tra1-FATKIN regions refine to nearly uniform resolution the Tra1-HEAT region showed a wider range of resolutions (and map quality), indicating remaining flexibility within this region. To further refine the Tra1-HEAT region, the core, and Tra1-FATKIN regions were subtracted from the particle images and the particle box was recentered around the Tra1-HEAT region. The resulting particles were subjected to alignment-free three-dimensional classification and a single good class showing the highest resolution features was selected, resulting in 285,738 particles. These particles were refined to 3.6 Å, and subjected to multibody refinement by masking the Tra1-HEAT into three parts (top, middle, and bottom – set of helices), which refine to 3.9, 3.4, and 3.7 Å. While the resolution of the Tra1-HEAT region did not improve, the map quality did in parts, allowing better interpretability of the peripheral regions. All resolution calculations were determined from gold-standard refinements at a Fourier Shell Correlation of 0.143 (*Scheres and Chen, 2012*; *Rosenthal and Henderson, 2003*).

Some of the software packages mentioned above were configured by SBgrid (*Morin et al., 2013*).

## Model building and refinement

The multibody maps for the core, Tra1-FATKIN, and Tra1-HEAT (top, middle and bottom) were combined for model building. The models for the Tra1 (PDB:5ojs) and Arp module (PDB:5i9e) (*Diaz-Santín et al., 2017*; *Cao et al., 2016*) were rigid-body docked into the combined map and adjusted where needed. The remaining density was manually built using COOT by building poly-alanine chains for all unaccounted density. Each chain

**Table 1.** Refinement table.

| Data collection, map and model refinement, validation | |
| --- | --- |
| **Data collection** | |
| Dataset | NuA4 |
| Microscope | Talos Arctica |
| Stage type | Autoloader |
| Voltage (kV) | 200 |
| Detector (mode) | Gatan K3 (super-resolution) |
| Pixel size (Å) | 1.117 (0.5585) |
| Electron dose (e$^-$/Å$^2$) | 38 |
| **Reconstruction** | |
| Software | RELION |
| Particles | 635,860 |
| Box size (pixels) | 320 |
| Map resolution (Å) | 3.1 |
| Map sharpening $B$-factor (Å$^2$) | −78 |
| **Coordinate refinement** | |
| Software | PHENIX |
| Resolution cutoff (Å) | 3.0 |
| FSC$_{model\text{-}vs\text{-}map}$ = 0.5 (Å) | 3.0 |
| Model-data fit (CC$_{mask}$) | 0.84 |
| **Model** | |
| Residues | |
| Protein | 5259 |
| *B*-Factors | |
| Protein | 65.28 |
| RMS deviations | |
| Bond lengths (Å) | 0.002 |
| Bond angles (°) | 0.507 |
| **Validation** | |
| Molprobity score | 1.72 (88th) |
| Molprobity clashscore | 6.88 (87th) |
| Rotamer outliers (%) | 0.02 |
| Cβ deviations (%) | 0 |
| Ramachandran plot | |
| Favored (%) | 95.03 |
| Allowed (%) | 4.97 |
| Outliers (%) | 0 |

segment was then inspected for stretches of high-quality density that could allow for the identification of potential side chain patterns that were then searched for within the sequences of known proteins within the complex (https://github.com/Stefan-Zukin/blobMapper; *Zukin, 2021*). Chain identification was also aided by secondary structure prediction and sequence conservation (*Ashkenazy et al., 2010*; *Ashkenazy et al., 2016*; *Buchan et al., 2013*; *Buchan and Jones, 2019*; *Jones and Cozzetto, 2015*). The resulting protein model was iteratively refined using PHENIX and manual adjustment in COOT (*Afonine et al., 2018b*; *Emsley et al., 2010*). The model was validated using MTRIAGE and MOLPROBITY within PHENIX (*Afonine et al., 2018a*; *Chen et al., 2010*). The refinement statistics are given in *Table 1* and show values typical for structures in this resolution range. The FSC curve between the model and the map shows good correlation up to 3.0-Å resolution according to the FSC = 0.5 criterion (*Rosenthal and Henderson, 2003*).

## dCypher-binding assays

dCypher-binding assays to PTM-defined histone peptides and semisynthetic nucleosomes were performed as previously (*Weinberg et al., 2019*; *Marunde et al., 2022b*; *Marunde et al., 2022a*; *Morgan et al., 2021*). Briefly, 5 µl of recombinant NuA4 HAT and TINTIN module Queries were titrated against 5 µl of histone peptide (100 nM) or nucleosome (10 nM) Targets and incubated for 30 min at room temperature in the appropriate optimized assay buffer Peptides: 50 mM Tris pH 7.5, 0.01% Tween-20, 0.01% BSA, 0.0004% poly-L-lysine, 1 mM TCEP; Nucleosomes: 20 mM Tris pH 7.5 [150 mM NaCl for HAT module or 200 mM NaCl for TINTIN module], 0.01% NP-40, 0.01% BSA, and 1 mM DTT [0.4 µg/ml sheared salmon sperm DNA (SalDNA) for HAT module or 0.04 µg/ml SalDNA for TINTIN module]. Then for the 6xHIS-tagged HAT module a 10 µl mixture of 5 µg/ml AlphaLISA Nickel chelate acceptors beads (*PerkinElmer #AL108*) and 10 µg/ml AlphaScreen donor beads (*PerkinElmer #6760002*); or for FLAG-tagged TINTIN module a 10 µl mixture of 1:400 anti-FLAG antibody (*MIlliporeSigma # F7425*), 5 µg/ml AlphaLISA protein-A acceptors beads (*PerkinElmer #AL101*), and 10 µg/ml donor beads were added, followed by a 60-min incubation. Alpha counts were measured on a *PerkinElmer 2104 EnVision* (680 nm laser excitation, 570 nm emission filter ± 50 nm bandwidth). To optimize salt (NaCl), competitor SalDNA concentrations, and query probing concentration, 2D assays were performed by titrating both query and salt (or SalDNA) against nucleosome substrates (Unmodified (WT), H3K4me3, H3K36me3, and biotin-DNA). It was found that the HAT module had weak DNA-binding ability that could be competed away with SalDNA (data not shown). Discovery screens consisting of 77 nucleosome substrates (*EpiCypher #16-9001*) were used to test the HAT (50 nM) and TINTIN (50 nM) modules (*Figure 3—figure supplement 5*). All binding interactions were performed in duplicate.

## Creation of figures

Depiction of molecular models were generated using PyMOL (The PyMOL Molecular Graphics System, version 1.8, Schrödinger), the UCSF Chimera package from the Computer Graphics Laboratory, University of California, San Francisco (supported by National Institutes of Health P41 RR-01081) and UCSF ChimeraX developed by the Resource for Biocomputing, Visualization, and Informatics at the University of California, San Francisco, with support from National Institutes of Health R01-GM129325 and the Office of Cyber Infrastructure and Computational Biology, National Institute of Allergy and Infectious Diseases (*Pettersen et al., 2004*; *Goddard et al., 2018*; *Pettersen et al., 2021*). Protein domains graphs (*Figures 2 and 3*, *Figure 1—figure supplement 2*) were generated using domainsGraph.py (https://github.com/avibpatel/domainsGraph; *Patel, 2022*). Crosslinking mass spectrometry figures were generated using Xlink Analyzer (*Kosinski et al., 2015*). ColabFold MSA and predicted alignment error figures were generated using ColabFold AlphaFold2-Advanced Notebook (https://colab.research.google.com/github/sokrypton/ColabFold/blob/main/beta/AlphaFold2_advanced.ipynb) (*Mirdita et al., 2022*).

## Acknowledgements

We thank Anthony Iavarone for mass spectrometry data collection and analysis, Patricia Grob, Daniel Toso, and Jonathan Remis for electron microscopy support, Abhiram Chintangal and Paul Tobias for computing support, Basil Greber, Vignesh Kasinath, and Michael-Christopher Keogh for feedback on manuscript. This work was funded through NIGMS grant R35-GM127018 to Eva Nogales, and

R44GM117683 (AlphaLISA assay development, aka. dCypher) and R44GM116584 (Nucleosome diversity) to EpiCypher. EN is a Howard Hughes Medical Institute Investigator.

## Additional information

### Competing interests

Matthew R Marunde, Irina K Popova: EpiCypher is a commercial developer and supplier of reagents (e.g. PTM-defined semi-synthetic nucleosomes; dNucs) and platforms (dCypher) used in this study. The other authors declare that no competing interests exist.

### Funding

| Funder | Grant reference number | Author |
|---|---|---|
| National Institute of General Medical Sciences | R35-GM127018 | Eva Nogales |
| National Institute of General Medical Sciences | R44GM117683 | Matthew R Marunde Irina K Popova |
| National Institute of General Medical Sciences | R44GM116584 | Matthew R Marunde Irina K Popova |
| Howard Hughes Medical Institute | | Eva Nogales |

The funders had no role in study design, data collection, and interpretation, or the decision to submit the work for publication.

### Author contributions

Stefan A Zukin, Matthew R Marunde, Investigation, Writing – review and editing; Irina K Popova, Katarzyna M Soczek, Investigation; Eva Nogales, Funding acquisition, Writing – review and editing; Avinash B Patel, Investigation, Writing – original draft

### Author ORCIDs

Eva Nogales ⓘ http://orcid.org/0000-0001-9816-3681
Avinash B Patel ⓘ http://orcid.org/0000-0001-9140-8375

### Decision letter and Author response

Decision letter https://doi.org/10.7554/eLife.81400.sa1
Author response https://doi.org/10.7554/eLife.81400.sa2

## Additional files

### Supplementary files

• MDAR checklist

### Data availability

The cryo-EM maps and coordinate models have been deposited in the Electron Microscopy Data Bank with the accession code EMD-28575 (NuA4 full), EMD-28563 (NuA4 core), EMD-28565 (NuA4 Tra1-FATKIN), EMD-28566 (Tra1-HEAT), EMD-28568 (Tra1-HEAT-top), EMD-28569 (Tra1-HEAT-middle), EMD-28567 (Tra1-HEAT-bottom) and in the Protein Data Bank with the accession code PDB-8ESC (NuA4). Plasmids for HAT and TINTIN expression have been made available through Addgene (Catalog #193325 (S.c. NuA4 HAT) and #193326 (S.c NuA4 TINTIN)).

The following datasets were generated:

| Author(s) | Year | Dataset title | Dataset URL | Database and Identifier |
|---|---|---|---|---|
| Patel AB, Zukin SA, Nogales E | 2022 | Structure of the Yeast NuA4 Histone Acetyltransferase Complex | https://www.rcsb.org/structure/8ESC | RCSB Protein Data Bank, 8ESC |
| Patel AB, Zukin SA, Marunde MR, Popova IK, Soczek KM, Nogales E | 2022 | Structure and Flexibility of the Yeast NuA4 Histone Acetyltransferase Complex | http://www.ebi.ac.uk/pdbe/entry/emdb/EMD-28575 | Electron Microscopy Data Bank, EMD-28575 |
| Patel AB, Zukin SA, Marunde MR, Popova IK, Soczek KM, Nogales E | 2022 | Structure and Flexibility of the Yeast NuA4 Histone Acetyltransferase Complex | http://www.ebi.ac.uk/pdbe/entry/emdb/EMD-28563 | Electron Microscopy Data Bank, EMD-28563 |
| Patel AB, Zukin SA, Marunde MR, Popova IK, Soczek KM, Nogales E | 2022 | Structure and Flexibility of the Yeast NuA4 Histone Acetyltransferase Complex | http://www.ebi.ac.uk/pdbe/entry/emdb/EMD-28565 | Electron Microscopy Data Bank, EMD-28565 |
| Patel AB, Zukin SA, Marunde MR, Popova IK, Soczek KM, Nogales E | 2022 | Structure and Flexibility of the Yeast NuA4 Histone Acetyltransferase Complex | http://www.ebi.ac.uk/pdbe/entry/emdb/EMD-28566 | Electron Microscopy Data Bank, EMD-28566 |
| Patel AB, Zukin SA, Marunde MR, Popova IK, Soczek KM, Nogales E | 2022 | Structure and Flexibility of the Yeast NuA4 Histone Acetyltransferase Complex | http://www.ebi.ac.uk/pdbe/entry/emdb/EMD-28568 | Electron Microscopy Data Bank, EMD-28568 |
| Patel AB, Zukin SA, Marunde MR, Popova IK, Soczek KM, Nogales E | 2022 | Structure and Flexibility of the Yeast NuA4 Histone Acetyltransferase Complex | http://www.ebi.ac.uk/pdbe/entry/emdb/EMD-28569 | Electron Microscopy Data Bank, EMD-28569 |
| Patel AB, Zukin SA, Marunde MR, Popova IK, Soczek KM, Nogales E | 2022 | Structure and Flexibility of the Yeast NuA4 Histone Acetyltransferase Complex | http://www.ebi.ac.uk/pdbe/entry/emdb/EMD-28567 | Electron Microscopy Data Bank, EMD-28567 |

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
