## [Editor Report]

This manuscript provides insights into the architecture of the yeast histone acetyltransferase complex NuA4 and is of broad interest to those studying transcription and chromatin modification. The cryo-EM data are of very high quality, and enable the authors to devise a structural model that is in much better agreement with biochemical data than previously reported models. This structure represents an important puzzle piece towards a molecular understanding of chromatin modification.

---

## [Decision Letter]

**Decision letter after peer review:**

Thank you for submitting your article "Structure and Flexibility of the Yeast NuA4 Histone Acetyltransferase Complex" for consideration by *eLife*. Your article has been reviewed by 2 peer reviewers, and the evaluation has been overseen by a Reviewing Editor and Volker Dötsch as the Senior Editor. The following individual involved in the review of your submission has agreed to reveal their identity: Hauke Hillen (Reviewer #1).

Essential revisions:

1) Stoichiometry

The authors should perform further characterisation of their sample to confirm it is indeed a single biochemical species as they state. Techniques such as AUC, intact mass spectrometry, or mass photometry approaches could help to answer this question.

2) Role of nucleosome modifications

The lack of binding to the tetra acetylated nucleosome is the incorrect control as no binding is shown for the tri-acetylation in the absence of H3K4me3. Furthermore, the situation may be more complex a negative result for tetra- or even tri- acetylation does not discount that there is acetylation binding, which may only occur within the context of H3K4me3. As such acetylation binding may be unable to compensate for the loss of H3K4me3 interaction.

Presumably, if the role of acetylation is simply to liberate the H3 tail as discussed acetylation would have no effect on HAT module binding on peptides as substrates rather than nucleosomes. Indeed, the authors suggest they have performed this experiment (Page 4 Line 167, data not shown). If the effect is simply due to neutralisation of charge, mutant nucleosomes with charge substitution at histone tail acetylation positions would also mimic this effect.

As possibly the most interesting biological conclusions from the manuscript (as highlighted in the abstract) further biochemistry on the TINTIN and HAT nucleosome interaction is required to bolster these modified nucleosome experiments to show these interactions outside of the dCypher screen.

3) XL-MS

Given the authors here have cross-linked their complex, and that the cross-linking experiments performed by another group (with the complex clearly behaving differently) were so central to the story and figures of this paper, were expecting to have seen some XL-MS done on the authors own complex, to further validate their own structure.

*Reviewer #1 (Recommendations for the authors):*

– Line 61: should be "TINTIN".

– Figure 1-3: I think it would be helpful if the authors indicated the designations for the submodules mentioned in the legend also in the Figure, as the focussed maps are also named accordingly.

– Line 127 and 143: The reference of Figure 3C does not seem appropriate here – do the authors mean Figure 3A?

– Lines 154-161: In this paragraph, the authors compare NuA4 with SAGA and present negative stain reconstructions of both. It is not clear to me why the authors determined negative stain reconstructions if higher resolution cryo-EM reconstructions are available for both NuA4 (this study) and SAGA (refs. 76,77). I think the reader would benefit here from a short sentence at the beginning of the paragraph stating (a) that negative stain reconstructions of NuA4 and SAGA were experimentally determined and (b) what the purpose of this experiment was.

– Line 156: Should be "3-4" instead of "3,4".

– Line 387: should be "Tra1-FATKIN".

– Line 757 (Figure 2A): I was confused by the fact that the coloring in the subpanels differs from the overview – is the pink part in the overview equivalent to the grey part in the subpanels? Maybe this could be clarified in the legend, or colored identically?

– Line 762 (Figure 2B): Do the colored bars above and below the protein primary sequence schematic indicate the same things as in Figures 1-2? If yes, please indicate in the legend, or if no please explain.

– Line 765: to illustrate the extent?

– Figure 3-1C: It would be helpful if the protein names were indicated in the according color next to the depicted models.

– Figure 3-3 is discussed and referenced in the main text before the dCypher binding assays. I would therefore suggest swapping the order of Figures 3-2 and 3-3.

*Reviewer #2 (Recommendations for the authors):*

1) Stoichiometry

The authors should perform further characterisation of their sample to confirm it is indeed a single biochemical species as they state. Techniques such as AUC, intact mass spectrometry, or mass photometry approaches could help to answer this question.

2) Role of nucleosome modifications

The lack of binding to the tetra acetylated nucleosome is the incorrect control as no binding is shown for the tri-acetylation in the absence of H3K4me3. Furthermore, the situation may be more complex a negative result for tetra- or even tri- acetylation does not discount that there is acetylation binding, which may only occur within the context of H3K4me3. As such acetylation binding may be unable to compensate for the loss of H3K4me3 interaction.

Presumably, if the role of acetylation is simply to liberate the H3 tail as discussed acetylation would have no effect on HAT module binding on peptides as substrates rather than nucleosomes. Indeed, the authors suggest they have performed this experiment (Page 4 Line 167, data not shown). If the effect is simply due to neutralisation of charge, mutant nucleosomes with charge substitution at histone tail acetylation positions would also mimic this effect.

As possibly the most interesting biological conclusions from the manuscript (as highlighted in the abstract) further biochemistry on the TINTIN and HAT nucleosome interaction is required to bolster these modified nucleosome experiments to show these interactions outside of the dCypher screen.

3) XL-MS

Given the authors here have cross-linked their complex, and that the cross-linking experiments performed by another group (with the complex clearly behaving differently) were so central to the story and figures of this paper, were expecting to have seen some XL-MS done on the authors own complex, to further validate their own structure.

---

## [Author Response]

Essential revisions:1) StoichiometryThe authors should perform further characterisation of their sample to confirm it is indeed a single biochemical species as they state. Techniques such as AUC, intact mass spectrometry, or mass photometry approaches could help to answer this question.

We have performed mass photometry and incorporated the data in the revised manuscript. However, the results did not confirm nor denied our original statement because of issues concerning the effect of the detergent needed to be present in the buffer. We mentioned this in the text. We have included the data in one of the supplementary figures, and slightly soften the language in the manuscript to reflect the uncertainty of these results. Please do notice that the presence or absence of a HAT species by itself is totally secondary to the structural and functional results presented in the manuscript.

2) Role of nucleosome modificationsThe lack of binding to the tetra acetylated nucleosome is the incorrect control as no binding is shown for the tri-acetylation in the absence of H3K4me3. Furthermore, the situation may be more complex a negative result for tetra- or even tri- acetylation does not discount that there is acetylation binding, which may only occur within the context of H3K4me3. As such acetylation binding may be unable to compensate for the loss of H3K4me3 interaction.

From the reviewers comments the issues seems to be that we stated that “acetylation itself is not recognized.” We concede that this is not the only possibility. We have corrected the statement and now say “acetylation on its own is poorly or not at all recognized.”

Presumably, if the role of acetylation is simply to liberate the H3 tail as discussed acetylation would have no effect on HAT module binding on peptides as substrates rather than nucleosomes. Indeed, the authors suggest they have performed this experiment (Page 4 Line 167, data not shown). If the effect is simply due to neutralisation of charge, mutant nucleosomes with charge substitution at histone tail acetylation positions would also mimic this effect.

We would presume the same. However, the peptide panel results did not translate quite as directly to the nucleosome panel results as we would have expected. Because of this, we focused on the more relevant substrate results: the nucleosome panel.

While we think a charge neutralization mutant would be a good idea to test if the increased affinity of the acetylated nucleosomes comes from H3 tail accessibility, such experiment goes beyond the scope of the paper, in which we concentrated on the structure and on the relative affinities of different nucleosome substrates. This idea of tail accessibility, which has been proposed in several other referenced papers, is an important one, but not our focus here.

As possibly the most interesting biological conclusions from the manuscript (as highlighted in the abstract) further biochemistry on the TINTIN and HAT nucleosome interaction is required to bolster these modified nucleosome experiments to show these interactions outside of the dCypher screen.

The dCypher system was used towards our goal of determining which are the modifications that the HAT and TINTIN module would prefer to bind to. The panel provided us with all the previously indicated modifications that these two modules were shown or implicated to bind to (H3K4me3 and H3K36me3) plus many more. As a result of this work, we were able to show that H3K4me3 is a recognized mark but not H3K36me3. Additionally, we saw that H3K4me3 and acetylation on H3 increases nucleosome binding over just H3K4me3 alone. While further HAT/TINTIN nucleosome binding experiments would bolster the findings of the dCypher assay, these would go beyond the scope of our paper. Again, it is important to emphasize at this stage that competing manuscripts on the structure of NuA4, the main component of this manuscript, are now under review or accepted, and that the delay that studies that are tangential and incremental to the present work will delay publication dramatically.

3) XL-MSGiven the authors here have cross-linked their complex, and that the cross-linking experiments performed by another group (with the complex clearly behaving differently) were so central to the story and figures of this paper, were expecting to have seen some XL-MS done on the authors own complex, to further validate their own structure.

We are a bit confused by this comment. We do not see how the complex is “clearly behaving differently.” We would argue that the observations by Setiaputra et al. agree extremely well with what we observe, as is qualitatively demonstrated by the fact that all of their mappable crosslinks are within expected distance ranges in our atomic model. It is the previously published structure by another group that does not fit the XL-MS data, and that is why we included the mapping onto both structures in the supplementary figure. Furthermore, in no way do we consider that the crosslinking mass spec was “central to the story and figures of this paper.” It only appears in two supplemental figures. The first one, as just described, simply demonstrates that our atomic model fits the existing crosslinking data, while that of Wang et al. does not. We would argue that the fact that the already published crosslinking mass spectrometry data came from a third lab makes it a most independent form of validation. Furthermore, at the resolution we have, such data is not required in any case (it is a valuable tool when the resolution of the map is >4.5 Å and you lack side chain information). Our second use of the published XL-MS data was to validate the AlphaFold models. Whether we use previous data or do redundant crosslinking experiments ourselves again would seem irrelevant when analyzing the *in silico* generated models.

Reviewer #1 (Recommendations for the authors):– Line 61: should be "TINTIN".

Corrected

– Figure 1-3: I think it would be helpful if the authors indicated the designations for the submodules mentioned in the legend also in the Figure, as the focussed maps are also named accordingly.

Added extra description in the figure caption.

– Line 127 and 143: The reference of Figure 3C does not seem appropriate here – do the authors mean Figure 3A?

Corrected

– Lines 154-161: In this paragraph, the authors compare NuA4 with SAGA and present negative stain reconstructions of both. It is not clear to me why the authors determined negative stain reconstructions if higher resolution cryo-EM reconstructions are available for both NuA4 (this study) and SAGA (refs. 76,77). I think the reader would benefit here from a short sentence at the beginning of the paragraph stating (a) that negative stain reconstructions of NuA4 and SAGA were experimentally determined and (b) what the purpose of this experiment was.

We added the following, “To directly compare the relative flexibility of the two complexes we performed negative stain electron microscopy of the two HATs.” Because we wanted to compare the relative flexibility of the chromatin binding modules of the two complexes, and since comparing maps from different methods or numbers of particles threshold differently we decided to do negative stain EM ourselves.

– Line 156: Should be "3-4" instead of "3,4".

Corrected.

– Line 387: should be "Tra1-FATKIN".

Could not find, possibly corrected during the editing process.

– Line 757 (Figure 2A): I was confused by the fact that the coloring in the subpanels differs from the overview – is the pink part in the overview equivalent to the grey part in the subpanels? Maybe this could be clarified in the legend, or colored identically?

The figure legend says: “Within Tra1, the pseudo-kinase domain is colored light blue, the FAT domain is colored pale yellow, and the HEAT domain is colored pink.” This coloring scheme was used to highlight the regions of Tra1 that other components of the complex were interaction with.

– Line 762 (Figure 2B): Do the colored bars above and below the protein primary sequence schematic indicate the same things as in Figures 1-2? If yes, please indicate in the legend, or if no please explain.

We have added the following: “Eaf1 domain map, as introduced in Figure 1-2.”

– Line 765: to illustrate the extent?

Corrected to “is proportional to.”

– Figure 3-1C: It would be helpful if the protein names were indicated in the according color next to the depicted models.

The following was added, “Models are colored based on the AlphaFold2-predicted interacting regions (as seen in panel B).”

– Figure 3-3 is discussed and referenced in the main text before the dCypher binding assays. I would therefore suggest swapping the order of Figures 3-2 and 3-3.

Thank you for pointing this out. We have corrected it.